# Exosomes: Emerging Modulators of Pancreatic Cancer Drug Resistance

**DOI:** 10.3390/cancers15194714

**Published:** 2023-09-25

**Authors:** Marzia Di Donato, Nicola Medici, Antimo Migliaccio, Gabriella Castoria, Pia Giovannelli

**Affiliations:** Department of Precision Medicine, University of Campania “L.Vanvitelli”, Via L. De Crecchio 7, 80138 Naples, Italy

**Keywords:** exosomes in pancreatic cancer, pancreatic cancer drug resistance, gemcitabine resistance

## Abstract

**Simple Summary:**

The high mortality of pancreatic cancer (PaC) is due to different reasons: a lack of specific symptoms, an unlikely diagnosis, therapies’ paucity, and drug resistance onset. For this reason, it is of paramount importance to develop new strategies to cure this incurable malignancy. Exosomes are secreted by all kinds of cells and used for intercellular communications. They are also used by cancer cells to induce drug resistance. Understanding how PaC cells use exosomes in drug resistance onset represents a supplemental weapon to cure PaC patients.

**Abstract:**

Pancreatic cancer (PaC) is one of the most lethal tumors worldwide, difficult to diagnose, and with inadequate therapeutical chances. The most used therapy is gemcitabine, alone or in combination with nanoparticle albumin-bound paclitaxel (nab-paclitaxel), and the multidrug FOLFIRINOX. Unfortunately, PaC develops resistance early, thus reducing the already poor life expectancy of patients. The mechanisms responsible for drug resistance are not fully elucidated, and exosomes seem to be actively involved in this phenomenon, thanks to their ability to transfer molecules regulating this process from drug-resistant to drug-sensitive PaC cells. These extracellular vesicles are released by both normal and cancer cells and seem to be essential mediators of intercellular communications, especially in cancer, where they are secreted at very high numbers. This review illustrates the role of exosomes in PaC drug resistance. This manuscript first provides an overview of the pharmacological approaches used in PaC and, in the last part, focuses on the mechanisms exploited by the exosomes released by cancer cells to induce drug resistance.

## 1. Introduction

Pancreatic cancer (PaC) is an intractable malignancy with a poor survival rate and is the seventh leading cause of cancer deaths worldwide (GLOBOCAN 2018) [1]. In the early stages, PaC patients are usually asymptomatic, and, upon cancer progression, heterogeneous, nonspecific symptoms can appear, making PaC hard to be diagnosed. The high mortality rate is, above all, linked to the lack of appropriate screening and diagnostic approaches but also to the aggressive clinical course of this pathology and the lack of effective therapies. Chemotherapy, radiotherapy, and surgery are the only therapeutical options actually available to improve the survival rate and relieve the patients’ symptoms [1]. Unfortunately, patients quickly develop resistance both to radio- and chemotherapy. The mechanisms responsible for chemoresistance are only in part known and studied and, while findings have been reported about the factors involved in the gemcitabine resistance, those responsible for the resistance to nanoparticle albumin-bound (nab)-paclitaxel or other used drugs or to radiotherapy are not explored enough [2]. 

Some of these mechanisms could rely on the exosomes. Exosomes are vesicles secreted by cancer cells. They play the most diverse roles in cancer, helping tumor progression. On the other side, they also could allow scientists and physicians to diagnose and cure cancer. Exosomes represent key markers for the diagnosis [3] and prognosis [4] of PaC and can also be used as targets in PaC therapy, especially for their recently discovered role in the development of drug resistance. For their characteristics, these vesicles are promising weapons to reduce PaC incidence and mortality, but, to date, there are many technical difficulties in their utilization. In this review, we will analyze the role of exosomes in the therapy escape of PaC, paying particular attention to the mechanisms directly involving exosomes in gemcitabine resistance. 

## 2. Pancreatic Cancer and Pharmacological Approaches

PaC is an underhanded pathology with a poor prognosis. The paucity of specific symptoms hinders the diagnosis, which is often made very late, when surgical resection is no longer possible. Depending on the extent of the disease, most clinicians use a four-tiered staging system: resectable, borderline resectable, locally advanced, and metastatic disease [5]. For each stage of PaC, specific pharmacological treatments are actionable. Gemcitabine in mono- or combo therapy with nab-paclitaxel, or nab-paclitaxel alone or with cisplatin and capecitabine, or even the multidrug FOLFIRINOX are prevalently used [6]. Besides these cytotoxic treatments, other drugs are usable to treat PaC, such as Trametinib, Palbociclib, Trastuzumab, Bevacizumab, Vismodegib, and PEGPH20 [7,8]. Some of these are studied in clinical trials and are already approved by the Food and Drug Administration (FDA). Unfortunately, their use is limited to some PaC subtypes or patients [7] that must have molecular and genetic testing to check if their mutations are specifically targeted by these therapies [7,8].

Both resectable and borderline resectable PaCs are treated with a postoperative systemic therapy setting, and different trials demonstrated that 6 months of multidrug FOLFIRINOX treatment is the most efficient therapy, increasing the median overall survival (MOS) from 35 to 54.4 months and the disease-free survival (DFS) from 12.8 to 21.6 months [6]. Anyway, gemcitabine, alone or combined with capecitabine, remains the treatment of choice for patients that do not tolerate multidrug FOLFIRINOX [6]. 

Locally advanced cancer differs from the first two subtypes for an extensive vascular involvement that precludes surgical resection. Depending on the performance status of the patient and their tolerance to the treatment, this kind of PaC is commonly treated with gemcitabine, with gemcitabine plus nab-paclitaxel, or with FOLFIRINOX [9,10,11]. The support of systemic chemotherapy (gemcitabine in the absence or presence of erlotinib) with radiotherapy does not absolutely prolong the patient’s OS [12]. 

Unfortunately, metastatic disease is diagnosed in at least 50% of PaC patients. At that stage, chemotherapy remains the only chance and it is used palliatively for cancer-related symptoms and to prolong the life expectancy. Even in this PaC clinical subtype, FOLFIRINOX is the most efficient first-line therapy, because it improves the OS from 6.8 to 11.1 months as compared with gemcitabine [13]. Nonetheless, even in this case, gemcitabine remains the elite treatment for debilitated patients. A first-line three-phase study, comparing gemcitabine versus gemcitabine plus-paclitaxel, demonstrated that patients treated with the combined therapy showed an OS of 8.7 months compared with the 6.7 months OS of patients treated with the monotherapy [14]. The therapy with Olaparib, a PARP inhibitor, is successfully used in BRCA1- and BRCA2-positive metastatic PaC [15]. This represents the first biomarker-based therapy in PaC. In metastatic PaC, gemcitabine is also used as second-line therapy in patients who had progressed on first-line FOLFIRINOX, while a combination of fluorouracil plus leucovorin with nano-liposomal irinotecan is used in patients who had progressed on first-line gemcitabine-based therapy [6,16]. 

Despite the significant number of therapeutical drugs used in PaC therapy, this cancer often develops resistance. The mechanisms leading to chemoresistance are various, ranging from genetic factors to the tumor microenvironment influence, without neglecting the most recent discoveries about the ability of exosomes in inducing drug resistance. This aspect will be extensively discussed in the subsequent sections of this review.

## 3. Exosomes Composition

Exosomes, a particular and specific group of extracellular vesicles (EVs), are released by different cell types, and much of the evidence indicates that cancer cells produce a higher number of exosomes as compared with normal cells [17,18,19]. These vesicles substantially differ in the formation process from the ectosomes, a second broad group of EVs [17]. While ectosomes are generated through the budding of the plasma membrane, exosomes are of endosomal origin and are initially produced as multivesicular bodies (MVBs), whose cargo is influenced by their interactions with intracellular organelles and vesicles [17,18,19]. Even if collected in one unique group, the exosome size and composition substantially vary. They show a diameter ranging from 30 to 150 nm and are made up of a lipidic bilayer-enveloping cytosol without organelles but enriched with protein and nucleic acids (Figure 1) [18]. 

The bilayer presents a lipidic part composed of sphingomyelin, cholesterol, phosphatidylserine, ceramide, and a protein fraction. In particular, the proteins expressed on the exosome membrane are responsible for the selection of target cells and the exosome uptake [20]. Different proteins are distributed inside and on the membrane surface of exosomes; some among them are specific types of exosomes and characteristics of the origin cells and tissues, while others are nonspecific types of exosome proteins present in all exosomes, regardless of the origin tissues. Specific types of exosome proteins are adhesion molecules (integrins, ephrin, and, mostly, the epithelial cell adhesion molecule—EPCAM), specific tetraspanins (CD9, CD63, and CD81), major histocompatibility complex (MHC) I and II class molecules and growth factors [19], and also heat shock, cytoskeleton, apoptotic, and cell signaling proteins. Nonspecific types of exosome proteins are, instead, components of the exosome biogenesis process, such as Rab GTPases (Rab2 and Rab7), ALIX, and flotillin [19]. 

Exosome content is also made up of different kinds of DNA and RNA molecules, including single- and double-stranded DNA (ssDNA and dsDNA); messenger, transfer, and ribosomal RNA (mRNA, tRNA, and rRNA); microRNA (miRNA); and non-coding RNA (ncRNA) [19,21]. In sum, the composition of exosomes mainly reflects the parental cells and origin tissue, except for the proteins involved in their biogenesis, which are shared by all the exosomes. 

Exosomes can transport the genomic and proteomic signatures typical of the tumor cells from which they derive. These unique signatures make exosomes ideal for cancer detection but, mainly, confer to exosomes derived from cancer cells the ability to promote cancer progression and transform healthy cells [22,23]. Oncogenes associated with various cancers can often be found in the exosomes secreted by tumor cells [22] as happens for KRAS^+^, present at a high concentration in exosomes from patients with PaC [24,25,26]. Another example is represented by EGFR [22,27] in PaC or by TGFa in colorectal cancer cells [22]. A peculiarity of the PaC-derived exosomes’ composition is the presence of Glypican-1 (GPC1), a cell surface proteoglycan. The cells of many cancers, particularly breast and pancreatic cancer, express high levels of this protein [28,29]. Melo and colleagues demonstrated that the levels of GPC1-positive circulating exosomes might be used to detect PaC at an early stage, with a sensitivity and a specificity higher than the level of the CA19-9 marker [3]. Another difference in the PaC-derived exosomes’ composition was highlighted by Tao and colleagues [30]. Their study shows that the lipid composition of the patients’ derived blood exosomes was different from the healthy controls and that variations in the PaC metabolites were reflected in the serum exosomes [30]. Their study shows that a group of 270 exosomal metabolic biomarkers, belonging to 20 lipidic species, was differentially expressed between the PaC and healthy subjects and exhibited a specific association with the patients’ clinicopathological features and PaC markers [30]. The lysophosphatidylcholine 22:0 and the [Plasmenyl-phosphatidylcholine 36:0; phosphatidylcholine (P-14:0/22:2)] are associated with CA19-9, CA 242, tumor stage, and lymphocytes count, while [PE (16:0/18:1)] is not only associated with the two PaC markers and the tumor stage but, mostly, is statistically correlated with the patients’ overall survival in PaC [30]. Other proteins are specifically expressed only on the exosomes released by PaC-derived cell lines and not on the exosome surface of normal cells. Among these are the extrachromosomal histone H2B; claudin 4 (CLDL4), involved in the tight junction formation and overexpressed in PaC; the epithelial protein EPCAM; and CD151, a tetraspanin family member implicated in cancer initiation and metastases [31]. 

## 4. Exosomes Biogenesis and Secretion

Exosomes are naturally and constitutively secreted into the extracellular space by both the healthy and sick cells of living beings, where they allow intercellular communication and take part in a wide variety of physiological or pathological processes, such as the immune response, neuronal communication, organ or cancer development, and growth. The formation process of PaC-derived exosomes is the same as the other cells and is tightly associated with the endocytic pathway (Figure 2).

The first step for exosome generation is the formation of an endocytic vesicle containing extracellular components and its fusion with an early endosome (EE) [18]. The EE follows two different fates: it might become recycling exosomes, thereby fusing with the plasma membrane and returning the cargo to it, or, after different maturation processes, it might differentiate into a late endosome (LE) and, lastly, into a multivesicular body (MVB) [19,21]. The MVB contains a certain number of intraluminal vesicles (ILVs), formed through the invagination of the membrane of the EEs, by the cargo sequestration and its distribution into the vesicles. ILVs are the effective exosome precursors [32] and their biogenesis is mainly regulated by different mechanisms, generally divided into Endosomal Sorting Complexes Required for Transport (ESCRT)-dependent and ESCRT-independent [18,19,21]. The ESCRT-mediated ILVs biogenesis depends on a sequence of processes coordinated by the four ESCRT or other support proteins such as clathrin, epsin-15, and Alix [19] and leads, through the membrane remodeling, to the vesicles sprouting. 

In the ESCRT-independent mechanisms, different molecules (tetraspanins, membrane sphingolipids, ceramides, the ceramide transport protein (CERT), and the Gi-coupled S1P1 receptor) take part in the ILV formation at various levels. Among them, tetraspanins are generally involved in more than one step of this process, such as the membrane compartmentalization into functional domains or cargo sorting [19,21]. Both sphingolipids and ceramide are particularly involved in membrane deformation, while CERT helps the ceramides to translocate from the Golgi and the endoplasmic reticulum to the endosomes, allowing for the membrane curvature, a local geometrical characteristic that typifies the membrane shape [33]. Recent studies have shown that Rab31 and S1P1 are also involved in ESCRT-independent exosome formation. Rab31 is a GTPase that, once activated, induces the membrane budding, while the Gi S1P1 receptor is involved in MVB maturation [21]. 

Before the formation of the vesicle, the cell picks out the appropriate cargo through a sorting process, different for each class of molecules carried by exosomes [17,21,32,34]. Monoubiquitination is the common tag to direct protein to the exosomes, even if this is not always true. Anyway, the monoubiquitinated proteins are recognized by different exosome markers containing ubiquitination recognition motifs, the better known being the Vps/27/Hrs protein, part of the ESCRT 0 complex, and the ESCRT I or II [35]. Before being included in exosomes, deubiquitinating enzymes remove the ubiquitin from the cargo proteins. Sometimes, ubiquitination is not sufficient for protein sorting. For instance, PD-L1 is delivered by Vps27/Hrs in new vesicles in the presence of ubiquitination but also of Vp27/Hrs phosphorylation by ERK [36]. Ubiquitination is not the only tag to address the proteins in ILVs. GPCRs can be directly recruited by Alix, another important exosome marker. Furthermore, other proteins are selected by a mechanism independent of ubiquitination as well as ESCRT [34,37]. These proteins possess a KFERQ domain and require the membrane protein LAMP2A that works by a mechanism dependent on the molecular chaperone HSC70, Alix, CD63, Syntenin-1, Rab31, and ceramides [37]. 

Exosomes also contain non-protein cargo, such as nucleic acids. The mechanism used to direct these molecules in exosomes is poorly understood and the scant published data show that it is highly selective, because specific proteins bind a reduced number of molecules. miRNA can be addressed to exosomes by binding specific sumoylated proteins such as heterogeneous nuclear ribonucleoprotein A2B1 (hnRNPA2B1) localized into the exosome membrane [38]. Mutant-KRAS is another protein able to alter the miRNA content in vesicles from colorectal cancer cells [39,40]. Tetraspanin 8 (Tspan8) is involved in the sorting of different miRNAs [41]. Anyway, there are no general mechanisms explaining how the RNA is loaded in vesicles, and the proteins involved in this process are collected in a generic group of the RNA-binding protein class of exosomal RNA sorting [42]. Otherwise, the mechanisms for the DNA sorting in exosomes are completely unknown and it is necessary to study this field for a major control of cancer behavior. 

When the production of the ILVs is completed, MVBs have two choices [19,34]. They can be degraded by lysosomes or fused with the plasma membrane to release the ILVs as exosomes. Different proteins expressed on the MVB membrane are responsible for the vesicle’s fate such as the cholesterol content, the expression of sphingosine-1 phosphate (S1P), a ceramide metabolite, or several Rab GTPases, such as Rab7, Rab31, and Rab27A and B. Conversely, Rab7 favors late endosome fusion with lysosomes, thereby reducing the exosomes secretion. Rab31 counteracts the Rab7 action and recruits the GTPase-activating protein TBC1D2B, thus increasing the exosomes release [43]. Rab27A and B are the most known mediators of exosome release. They ensure the right tethers between the SNARE proteins localized on vesicles (v-SNARE) as well as the target membrane (T-SNARE), allowing the SNARE complex formation, the completion of the fusion process, and the exosome release [19]. 

Once released, exosomes enter the target cells through different mechanisms that are still not clear. They can directly fuse with the plasma membrane of the target cells or undergo different types of endocytosis (clathrin- or caveolin- or lipid raft-mediated endocytosis, pinocytosis, or phagocytosis) or directly interact with the plasma membrane receptors such as PD-L1, TRAIL, FasL, and TNF located on the membrane of cancer cells [44,45]. However, the most used mechanism for exosome uptake is endocytosis. Once in the cell, the cargo can carry out its function.

## 5. Role of Exosomes in PaC

Because of their ability to carry a multitude of different molecules, exosomes are involved in distinct processes of cancer biology (Figure 3).

The role of exosomes in PaC is not quite different from exosomes acting in other cancers. As such, they alter and control cancer progression by influencing proliferation, migration, invasion, immunoregulation, and chemoresistance, both at local and systemic levels. As in other cancers, exosomes are used in the diagnosis and prognosis of PaC. Moreover, they play an exclusive role in the pathogenesis of diabetes. 

### 5.1. Exosomes in PaC Progression

Exosomes influence PaC progression and metastases’ formation by altering the proliferation, apoptosis, migration, invasiveness, and metabolism of its cells [41]. 

Cancer growth is positively or negatively controlled by exosomes. The data in the literature demonstrate that different elements of the PaC-derived exosomal cargo control PaC proliferation. In highly malignant pancreatic PC-1.0 cells, the zinc ion transporter protein 4, ZINC4, is the most upregulated exosome protein and strongly enhances cell proliferation in in vivo and in vitro models [46]. By regulating the expression and the localization of p27, miRNA-222, contained in tumor-secreted exosomes, promotes proliferation and invasion of pancreas cancer and nearby cells [47]. 

A plethora of proteins, miRNAs, and other signaling elements, able to alter cell proliferation, can be loaded into exosomes and exert their action on neighboring cancer cells and on the tumor stroma that plays a pivotal role in PaC progression and metastases. The PaC microenvironment is highly heterogeneous and contains fibroblasts (cancer-associated fibroblasts—CAFs), adipocytes, immune cells, tumor-associated macrophages (TAM), and pancreatic stellate cells (PSC). These elements are all surrounded by the extracellular matrix (ECM) [48]. The behavior of cancer is, therefore, strongly influenced by the interactions between cancer cells and the microenvironment. In this context, through exosomes, cells at distant sites strongly communicate [41,49]. Cancer cell proliferation is often induced by exosomes derived from cancer-associated stroma. Exosomes released by gemcitabine-treated CAFs, for instance, increase the PaC proliferation and chemoresistance by upregulating the Snail transcription factor [50]. In a nutrient-depleted context, CAF-derived exosomes also transfer metabolic substrates to PaC-derived MiaPaCa-2 and BxPC3 cell lines, thus restoring cell proliferation [51]. Again, PSCs release exosomes triggering PaC proliferation, migration, and chemokine expression. GW4896, an exosome release inhibitor, prevents these effects. The exosome cargo includes different miRNAs, such as miR-21-5p, oncomiR, miR-1246, and miR-1290. Moreover, PaC cells release exosomes influencing the PSC activation and the profibrogenic activity by inducing the production of the procollagen type I C peptide and the mRNA expression of fibrosis-related genes and of alpha-smooth muscle actin (ACTA2) [52]. In this way, PaC-derived exosomes contribute to developing a permissive microenvironment for PaC progression [52].

Natural killer (NK)-derived exosomes possess the ability to inhibit the malignant transformation of PaC cells. Co-cultured with NK cells, it inhibits the proliferation, migration, survival, and viability of PaC cells by reducing the IL-27 level [53]. 

By inducing apoptosis, exosomes from both PaC and stroma cells might exert a negative action on tumor growth. Exosomal nanoparticles secreted by cancer cells induce the mitochondria-dependent apoptotic pathway, through the activation of the pro-apoptotic phosphatase and tensin homolog deleted on chromosome 10 (PTEN) as well as glucose synthase kinase-3β (GSK-3β) [54]. 

MiR-145-enriched exosomes by tumor-associated stroma (TAS) play a tumor-suppressive role on PaC-derived cells [55]. 

Exosomes derived from cancer or tumor-associated cells might also alter the migratory and metastatic potential of PaC cells. 

Exosomes from highly invasive pancreatic cancer-derived PC-1.0 cells increase the invasiveness of weakly metastatic PC-1 cells [56]. These exosomes contain at least 62 different upregulated miRNAs and, among them, miR-125b-5p most significantly promotes the aggressive phenotype in less malignant PC-1 cells. Furthermore, this miRNA is upregulated in highly invasive PaC cells, likely for MEK/ERK signaling activation. The STARD13 oncosuppressor is one of the miR-125-5p targets and its expression is negatively related to PaC prognosis [57]. Similarly, miR-887-3p represents another miRNA responsible for cancer progression, through the STARD13 downregulation. By targeting STARD13, this miRNA increases cell proliferation, reduces apoptosis, and fosters aggressiveness, thus promoting PaC progression [57].

### 5.2. Exosomes in Pathogenesis of Cancer-Associated Diabetes

The peculiarity of PaC-derived exosomes is their role in the pathogenesis of diabetes often associated with PaC [58,59,60,61]. The mechanisms involved in this process are not fully understood, but several data have shown this correlation in PaC patients [62,63,64]. First, exosomes from PaC enter the skeletal muscle cells where they promote lipidosis while inhibiting glucose transporters [63]. Different miRNAs are involved in this process and some of them, such as miR-666-3p, miR-125-5p, and miR540-3p, promote the expression of FOXO1, a critical player in skeletal muscle insulin resistance. Others, such as miR-883b-5p and miR-151-3p, are responsible for Glut4 downregulation [63]. Second, the correlation between PaC and diabetes was discovered in PaC patients with newly diagnosed diabetes and analyzed by “in vitro” experiments in PaC-derived cell lines [64]. Diabetic PaC patients showed low levels of the glucose-dependent insulinotropic peptide (GIP), and exosomes from PaC cells inhibit insulin secretion in target cells by reducing both GIP and glucagon-like peptide-1 (GLP-1) [64]. These altered protein levels seem to be caused by different miRNAs (miR-197-3p, miR-4750-3p, miR6763-5p, and miR6796-3p) that suppress the expression of proprotein convertase subtilisin/kexin type 1/3 (PCSK1/3), thereby inhibiting GIP and GLP-1 [64].

These data confirm that the diabetes onset in PaC patients is strictly related to this kind of cancer and is due to different molecules produced by cancer cells, most of them transferred by exosomes.

## 6. Exosomes and PaC Drug Resistance

Besides the late diagnosis, one of the major plagues of PaC is the onset of chemoresistance. This attitude is due to the high heterogeneity of genetic mutations and the complex and dense stroma environment [65]. Gemcitabine is the most used chemotherapeutic drug and, consequently, the mechanisms leading to its resistance are well studied. It might depend on genetic pressures during PaC progression as well as the stroma-derived non-coding miRNA, proteins, or miRNAs involved in the Epithelial-to-Mesenchymal transition (EMT) [66,67] (Figure 4). 

The role of exosomes in cancer growth and spreading, as well as diagnostic and therapeutic factors, is now well studied and known enough, while there is less information about their involvement in drug resistance. 

Exosomes might contribute in two different ways to PaC resistance. They directly work by kicking out drugs from cancer cells, or indirectly act by delivering miRNA or mutated or overexpressed proteins in recipient drug-sensitive cells [19]. In the last case, the mechanisms of chemoresistance induced by exosomes in target cells are the same observed in drug-resistant cancer cells from which exosomes spring [19,65,68,69].

Because of the vesicular trafficking, there is an exchange of promoting chemoresistance factors between drug-resistant and drug-sensitive cells but also between cancer and microenvironment cells. One of the most basic mechanisms by which exosomes are actively involved in chemoresistance is the removal of gemcitabine from PaC cells and their microenvironment.

### 6.1. Exosomal-Delivered Proteins and Chemoresistance

A study by Muralidharan-Chari and colleagues has shown a direct correlation between the amount of released microvesicles and drug resistance. The authors observed that in different gemcitabine-treated pancreatic cancer cell lines, the amount and the variety in the size of the secreted vesicles are directly related to drug resistance. The gemcitabine treatment regulates the level of influx and efflux proteins on the vesicle membrane. While the influx proteins (ENT1) increase only in vesicles from gemcitabine-treated cells, the efflux proteins (MRP1, MRP5, and P-gp) are more copious only in vesicles from untreated cells, thereby indicating that vesicles from gemcitabine-treated cells are organized to trap the drug inside [70]. 

Zhao and colleagues have shown that gemcitabine resistance in PaC is related to the mutation of more than 165 genes [71] encoding for proteins involved in the most different functions, such as cell cycle and apoptosis, invasiveness, and antioxidant activities. A lot of these proteins, expressed in drug-resistant PaC cells, are loaded in exosomes and transferred to drug-sensitive cells within the same cancer. The activities of different proteins such as drug efflux pumps are linked to PaC chemoresistance [72,73,74]. These transporters include multidrug resistance-associated proteins (MRP) [73] such as MRP1 (also called ABCC1), some proteins of the ATP-binding cassette (ABC) superfamily such as p-glycoprotein (P-gp, ABCB1, or MDR1) [75], and the mitoxantrone resistance protein (MXR, or ABCG2, the breast cancer resistance protein). In such a way, specific inhibitors could be used as PaC therapy to restore or increase gemcitabine sensitivity [74]. The ABCG2 level increases in exosomes from pancreatic cancer cells with depleted GAIP-interacting protein C terminus (GIPC) [75]. GIPC modulates PaC cells’ autophagy, controls exosome biogenesis, and influences exosome content. GIPC depletion in PaC-derived cells increases the gemcitabine sensitivity but induces the production of ABCG2-enriched exosomes that are delivered to drug-sensitive cells where they can regulate the drug efflux to induce the chemoresistance [75]. 

Another protein transferred through exosomes is metalloprotease 14 (MMP14) [76]. This enzyme is one of the most frequently transferred proteins from gemcitabine-resistant PaC cells to sensitive ones. In recipient cells, MMP14 enhances drug resistance and promotes migration and sphere formation ability; thereby, it could be a good therapeutic option to target MMP14 in PaC exosomes to reduce drug resistance [76]. 

The gemcitabine resistance is also controlled by the exosomal transfer of ROS detoxification enzymes [69]. Gemcitabine is an antimetabolite and exerts its antiproliferative action by targeting ribonucleotide reductase (RR) and interfering with DNA synthesis [77]. After its entry into cancer cells, it is, first of all, phosphorylated by deoxycytidine kinase (dCK) and, subsequently, undergoes a series of phosphorylation steps that transform it into gemcitabine di- and tri-phosphate (dFdCDP and dFdCTP) [78]. dFdCDP inhibits RR, while dFdCTP is incorporated in both DNA and RNA [77,78]. In PaC cells, gemcitabine treatment induces the release of reactive oxygen species (ROS) [79], reinforcing the antiproliferative drug action. Gene expression analysis of PaC cells treated with exosomes from gemcitabine-treated cells reveals the presence of high levels of ROS detoxification enzymes, such as superoxide dismutase 2 (SOD2) and catalase (CAT), and the downregulation of dCK, the gemcitabine-metabolizing enzyme. In this way, conditioned media from gemcitabine-treated cells confer chemoresistance to PaC cells. Specifically, this ability is exclusive to the exosomes’ fraction because they contain both the SOD and CAT transcripts and miR-155, able to downregulate the dCK [69]. 

Another protein transferred by exosomes and able to confer chemoresistance to recipient cells is the Ephrin Type-A receptor2 (EphA2). Fan and colleagues demonstrated that exosomes isolated from the chemo-resistant PaC-derived PANC-1 cell line confer drug resistance to the other two PaC cell lines, MIA PaCa-2, and BxPC-3 with a lower gemcitabine resistance level. EphA2 is mainly loaded in the PANC-1 exosomes’ membrane because only this cell line overexpresses this receptor [68]. The mechanism by which EphA2 induces drug resistance in gemcitabine-sensitive cells is not yet clear, and the unique evident fact is that only the membrane-carried EphA2 contributes to the chemoresistance [68,80].

Impaired apoptosis is another cause of gemcitabine resistance [81]. Different factors, belonging to the Inhibitors of Apoptosis (IAP) family, such as XIAP, cIAP1, cIAP2, and survivin, are upregulated in PaC and their overexpression is negatively associated with drug sensitivity. Exosomes from PaC cells, acting on the tumor microenvironment (TME), are enriched with survivin, which works as a “multipurpose protein” [82], as it might promote the growth, metastases, survival, and chemoresistance of cancer cells [83]. Furthermore, extracellular survival decreases the drug sensitivity of TME cells [82]. 

### 6.2. Exosomal-Delivered miRNA and Chemoresistance

miRNAs are one of the most studied elements delivered by exosomes. They influence a plethora of cell processes involved in cancer’s onset, progression, and aggressiveness. Several miRNAs alter the drug sensitivity of cancer cells and, among them, miR-155, miR-210, miR-146a, and miR-365 are abundant in exosomes and affect the drug resistance of PaC [47,56,57,69].

As previously mentioned, exosomes from gemcitabine-treated PaC cells induced drug resistance by detoxifying the recipient cells of gemcitabine and gemcitabine-induced ROS. In the first action, miR-155 reduces the dCK levels by inhibiting the gemcitabine metabolism and action on nucleic acids [69]. Another study demonstrated that gemcitabine exposure increases the miR155 expression in PaC cells. The high level of this miRNA predisposes cells to gemcitabine resistance and exosome secretion. Finally, the miR-155 entry in recipient PaC cells induces chemoresistance by increasing anti-apoptotic activities [84].

miR-210 is another miRNA delivered by exosomes from cancer stem cells (CSCs) and is responsible for drug resistance in in vitro and in vivo models of PaC. It increases chemoresistance by inhibiting apoptosis and promoting the cell cycle through the mTOR pathway activation [67].

Recently, different studies observed that PaC acquires gemcitabine resistance through the expression of EMT markers, although the molecular mechanism linking these two processes is not yet fully elucidated. EMT consists of a phenotypical switch of cancer cells occurring by the expression of different proteins such as Snail, Slug, Twist, and Zeb1, thereby increasing the cell invasiveness. The Slug knockdown increased the CD133^+^ PaC cells’ drug sensitivity [66], while the EMT was positively related to gemcitabine resistance in PaC mouse models [85]. The tumor microenvironment of PaC is structurally peculiar. Desmoplasia is its main feature, while cancer-associated fibroblasts (CAFs) originating from pancreatic stellate cells (PCSs) are the most representative cells [48]. CAF-derived exosomes are enriched in Snail and miR-146a and promote EMT, metastases, and gemcitabine resistance in PaC cells. The use of GW4869, which suppresses the exosome secretion, reduces PaC survival after gemcitabine administration [50]. 

Even tumor-associated macrophages (TAM) contribute to PaC drug resistance in vitro and in vivo. These immune cells are components of the PaC microenvironment and, through their exosomes, transfer the miR-365 to gemcitabine-sensitive cancer cells by developing their chemoresistance [86]. miR-365 impairs the gemcitabine activation by increasing the intracellular concentration of triphosphate-nucleotides (NTP), which competes with dCK-phosphorylated gemcitabine (dFdCTP) for DNA incorporation [86].

In sum, these results highlight the new role of exosomes in promoting PaC chemoresistance (most of them are summarized in Table 1) and suggest that the combined use of the inhibitors of vesicles formation together with the classical chemotherapeutic agents might enhance the therapy efficiency. However, additional studies are needed in this direction.

## 7. Concluding Remarks

PaC is going to become the second leading cause of cancer death in 2030 in the Western world, mainly because of the increase in its incidence and the delay in diagnosis as well as its intractability and resistance onset. These aspects, together with the scant therapeutical options, make this cancer highly aggressive and clinically unactionable.

Exosome-induced resistance is an event that has only recently been explored, but the published results give a good chance to improve the efficiency of canonical therapies and develop new strategies to reduce the high PaC mortality. In particular, the identification of specific molecules carried by exosomes can allow for the development of new drugs or strategies to bypass chemoresistance. Furthermore, the specific biomolecular features (i.e., lipidic and protein composition) and the copious production of exosomes by cancer cells compared to normal cells make the inhibition of the exosomes’ release a simple and specific target for curing PaC.

The published data also describe how exosomes can induce resistance to other drugs, such as nab-paclitaxel. Nevertheless, these findings are mainly based on our knowledge of chemoresistance mechanisms studied in single cells rather than by directly analyzing the exosomes’ functions. For these and other reasons, the study of exosomes in PaC still represents an unexplored field that deserves further investigation.

PaC remains one of the leading causes of death worldwide. Despite some advancements that have been made in its treatment, the current therapies (mainly chemotherapy and targeted therapies) still have several drawbacks, such as the very limited strength, the severe side effects, and, not least, the elevated costs, which are generally due to the high rate of failure of the tested molecules. The implementation of studies in PaC cells, the surrounding microenvironment, and their derived exosomes might provide new hints for the identification of prognostic/predictive signatures, with important implications for drug screening, the clinical stratification of patients, and public health.

## Figures and Tables

**Figure 1 cancers-15-04714-f001:**
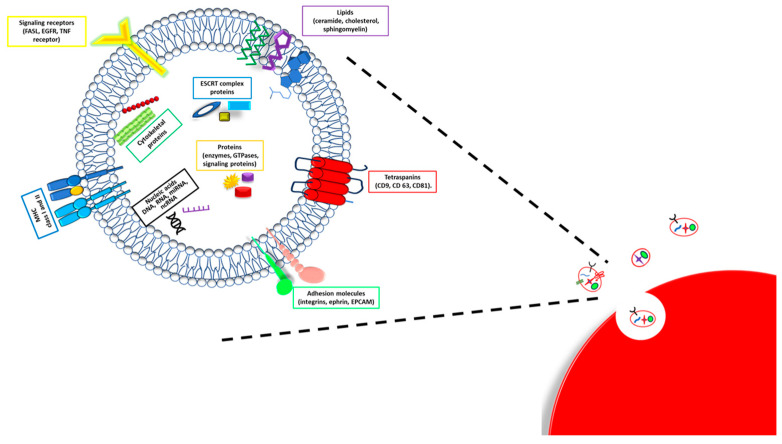
Exosome composition. A schematic illustration of exosome structure and composition.

**Figure 2 cancers-15-04714-f002:**
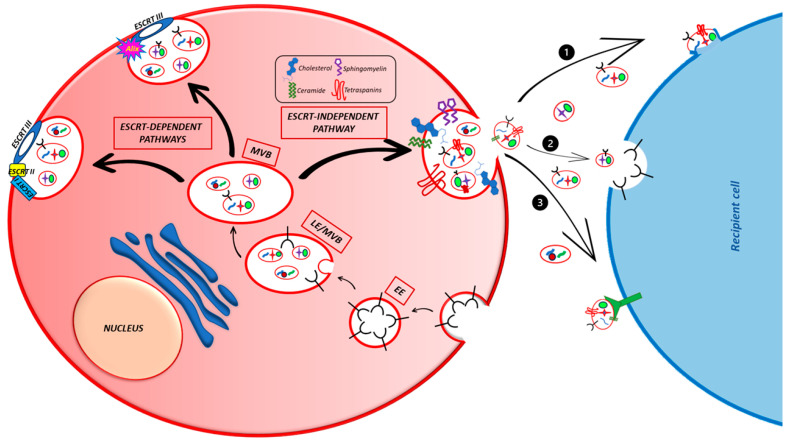
Principal mechanisms of exosome formation in cells. The figure illustrates the ESCRT-dependent and -independent pathways involved in exosome formation and the three ways used by exosomes to enter the target cells. (1) Direct fusion with the plasma membrane; (2) endocytosis; (3) direct interaction with plasma membrane receptors.

**Figure 3 cancers-15-04714-f003:**
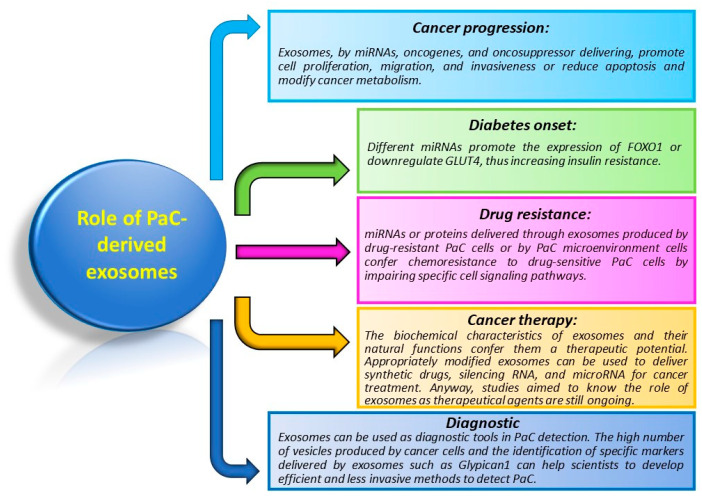
Principal functions of exosomes in PaC. The figure briefly resumes functions and mechanisms controlled by exosomes in PaC.

**Figure 4 cancers-15-04714-f004:**
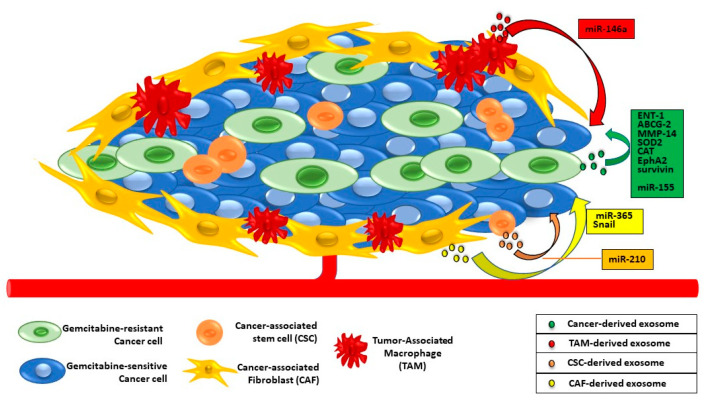
Exosomal trafficking between cancer and TME cells and gemcitabine resistance in PaC cells. Exosomes from drug-resistant PaC cells and/or from tumor-associated macrophages, fibroblasts, and stem cells contain proteins or miRNAs able to confer chemoresistance to gemcitabine-sensitive PaC cells.

**Table 1 cancers-15-04714-t001:** The table briefly outlines the most important and known exosomes-related regulators of gemcitabine resistance in PaC cells and their mechanism of action.

** *Exosomal-delivered proteins transfer* **	** *ROS detoxification enzymes* ** ** *(SOD2 and CAT)* **	It inhibits gemcitabine metabolites formation (dFdCDP and dFdCTP), responsible for drug action (by interfering with DNA synthesis and cancer cell proliferation) [69].
	** *MMP14* **	It enhances drug resistance and promotes migration and cell stemness [76].
	** *IAP* **	It inhibits the apoptosis in PaC cells [83].
	** *ABCG2* **	It triggers the gemcitabine drug efflux outside the PaC cells [75].
** *Exosomal-delivered* ** ** *miRNA* **	** *miRNA 155* **	It reduces the gemcitabine metabolism by downregulating the deoxycytidine kinase (dCK) levels [69,84].
	** *miRNA 210* **	It increases chemoresistance by inhibiting apoptosis and promoting the cell cycle through the mTOR pathway activation [67].
	** *miRNA 365* **	It impairs the gemcitabine activation by increasing the intracellular concentration of triphosphate-nucleotides (NTP), which competes with dFdCTP for DNA incorporation [86].

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
