# Peer review of "Exosomes: Emerging Modulators of Pancreatic Cancer Drug Resistance"

_cancers, 2023, doi:10.3390/cancers15194714_

Round 1
Reviewer 1 Report
The review describes therapeutic approaches that are currently used for therapy of pancreatic cancer (PaC) and the role of the exosomes in carcinogenesis and pathogenesis of PaC, including its chemoresistance. In general, the review is well-written and requires minor editing of English language.
I have the following suggestions and concerns regarding this manuscript.
1) nab-paclitaxel (line 10) should be defined as the nano-patricle albumin bound paclitaxel.
2) The authors have to show the precise names of the drugs used for PaC therapy (lines 59-60). The named them as the "other drugs" which is not appropriate.
3) Similarly, it's an important to show the unique features of patient-derived exosomes that are different from exosomes derived form healthy subjects, as was referred in 24 (the lines 134-135 and 138-139, as well).
4) The authors included ABCG2 transporter to the MRP-based family, which is not correct. MRP-related proteins differ from ABCG2 family. Similarly MRP inhibitors differ from the chemicals that block the activity of ABCG2 transporter (lines 360-261, ref 71)
5) I suggest to supplement the manuscript with the Table which illustrates the most potent and well-described exosome-related regulators of PaC chemoresistance with a brief description of their molecular mode of action.
The manuscript requires minor editing of English Language.
Author Response
Reviewer #1
The review describes therapeutic approaches that are currently used for therapy of pancreatic cancer (PaC) and the role of the exosomes in carcinogenesis and pathogenesis of PaC, including its chemoresistance. In general, the review is well-written and requires minor editing of English language.
I have the following suggestions and concerns regarding this manuscript.
1) nab-paclitaxel (line 10) should be defined as the nano-particle albumin bound paclitaxel.
We sincerely thank the reviewer for this clarification. We modified as suggested (lines 19).
2) The authors have to show the precise names of the drugs used for PaC therapy (lines 59-60). The named them as the "other drugs" which is not appropriate.
We now added the new drugs (lines 67-68).
3) Similarly, it's an important to show the unique features of patient-derived exosomes that are different from exosomes derived from healthy subjects, as was referred in 24 (the lines 134-135 and 138-139, as well).
We agree with the reviewer's observation and have scrutinized this matter. Please see lines 146-175.
4) The authors included ABCG2 transporter to the MRP-based family, which is not correct. MRP-related proteins differ from ABCG2 family. Similarly, MRP inhibitors differ from the chemicals that block the activity of ABCG2 transporter (lines 360-261, ref 71).
We apologize with the reviewer and modify this mistake (lines 397-414).
5) I suggest to supplement the manuscript with the Table which illustrates the most potent and well-described exosome-related regulators of PaC chemoresistance with a brief description of their molecular mode of action.
We accepted the reviewer suggestion and added a Tab1 (Table1) that briefly resume the most important and known exosomes-related regulators of Gemcitabine resistance in PaC cells and their mechanism of action.

Reviewer 2 Report
A review by Di Donato et al discusses a role of exosomes in drug resistance of pancreatic cancer cells.
Specific comments:
1. English should be revised throughout the manuscript.
2. Abstract should be written in more convincing and more scientific manner.
3. The Authors highlight a few times in the manuscript that "exosomes are secreted by cancer cells". However, it should be also mentioned that normal cells and non-cancerous cells also release exosomes.
4. Paragraphs 3 and 4 will benefit from figure(s).
5. Figure 2 - letter can be substantially enlarged as there are a lot of free space.
requires revision
Author Response
Reviewer #2
A review by Di Donato et al discusses a role of exosomes in drug resistance of pancreatic cancer cells.
We thank the reviewer for his/her time. In this version, we modified the different critical points as suggested.
Specific comments:
- English should be revised throughout the manuscript.
We modified the English language through the entire manuscript.
- Abstract should be written in more convincing and more scientific manner.
In this version, we completely modified the Abstract.
- The Authors highlight a few times in the manuscript that "exosomes are secreted by cancer cells". However, it should be also mentioned that normal cells and non-cancerous cells also release exosomes.
We agree with the reviewer. In this version we better explain this concept (lines 23-24; 110-112; 178-183).
- Paragraphs 3 and 4 will benefit from figure(s).
We added two new figures (Fig 1 and 2) in this version.
- Figure 2 - letter can be substantially enlarged as there are a lot of free space.
We modified it as indicated by the reviewer.

Round 2
Reviewer 2 Report
Comments have been addressed. However, I still recommend to enlarge the content of the figures.